# The mediating role of positive orientation in the relationship between personality traits and attitudes towards artificial intelligence

Wiesław Talik ⍟*

Institute of Sociological Science, John Paul II Catholic University of Lublin, Lublin, Poland

* talik@kul.pl

## Abstract

This study examines the mediating role of the positive orientation relationship between personality traits (extraversion, neuroticism, openness to experience, agreeableness, and conscientiousness) and attitudes towards artificial intelligence. The study was conducted in Poland with a sample of $N = 922$ adults. The research employed the Artificial Intelligence Attitude Scale (AIAS-4)**,** the Ten Item Personality Inventory (TIPI) and the Positivity Scale. The mediating effect of positive orientation on the relationship between personality traits and attitudes towards artificial intelligence was partially confirmed. The findings indicate that Big Five personality traits (with the exception of openness to experience) may lead to more positive attitudes towards AI and that this relationship is mediated by positive orientation. This research elucidates the mediating role of positive orientation, thereby contributing to a deeper understanding of the psychological processes driving individuals' attitudes towards AI. The insights gained can inform policymakers, technologists, and practitioners aiming to promote the widespread acceptance and adoption of AI technologies in society.

## Introduction

Artificial intelligence (AI) has become an integral part of our daily lives, influencing a wide range of areas, including recommendation algorithms, large language models (LLMs), medicine, home technologies, and autonomous vehicles. AI has emerged as a transformative force, reshaping industries, economies, and everyday lives. As AI technologies continue to develop rapidly, understanding how people perceive and interact with them becomes increasingly important [1].

Negative attitudes toward artificial intelligence (AI) can have far-reaching consequences for technology adoption and societal progress. Research shows that distrust or skepticism toward AI may reduce individuals' willingness to use AI-based tools, hinder the implementation of innovative systems, and limit potential benefits in areas such as healthcare, education, and public administration [2,3]. Such resistance can

**Data availability statement:** The study materials, data used for this article can be accessed at https://osf.io/yjegk/?view_only=366af2ca999b-43448ba9ead13f1a8ec1.

**Funding:** This work was supported by the Institute of Sociological Science, John Paul II Catholic University of Lublin (Grant No. 1118, received by WT). The funder had no role in study design, data collection and analysis, decision to publish, or preparation of the manuscript.

**Competing interests:** The author has declared that no competing interests exist.

slow organizational efficiency and exacerbate existing inequalities by widening the digital divide between those who adopt AI and those who reject it. Moreover, fear of job displacement and perceived threats to privacy or autonomy may intensify negative affect toward AI, further reducing engagement with its applications [4].

However, excessively positive and uncritical attitudes toward AI also carry significant risks. Studies on automation bias demonstrate that overreliance on algorithmic recommendations can impair human judgment, leading to the acceptance of incorrect outputs without sufficient verification [5]. Uncritical acceptance of AI outputs may also facilitate the perpetuation of embedded biases, resulting in unfair or discriminatory outcomes in domains like recruitment or criminal justice [6]. These findings underscore the need for balanced, informed attitudes toward AI that combine openness to innovation with critical evaluation of potential limitations and harms.

Personality traits have been the subject of extensive investigation for many years, with the objective of elucidating the manner in which individual differences in personality affect various aspects of life, including technology adoption. Recent research has indicated that personality traits play a pivotal role in shaping individuals' attitudes toward AI adoption, interaction, and trust. The Big Five personality traits have been instrumental in elucidating the variability in attitudes towards technological innovations, including AI [7–10].

A review of the literature on the relationship between personality traits and attitudes towards artificial intelligence (AI) has identified several key findings. Introverts exhibit more positive attitudes toward AI, whereas those high in conscientiousness and agreeableness are more forgiving of negative aspects. Furthermore, general trust leads to positive views of AI benefits [7]. Park and Woo [11] demonstrated that extraversion, agreeableness, conscientiousness, neuroticism, and personal innovativeness in information technology are related to various attitudes towards AI. Ammer et al. [12] reported a positive correlation between four personality traits (agreeableness, conscientiousness, extraversion, and openness) and a negative correlation between neuroticism. Sindermann et al. [13] emphasised the significance of openness to experience, extraversion, introversion, and neuroticism in shaping attitudes towards AI, laying the groundwork for exploring the role of personality traits in attitudes towards technology. Zhang [14] examines public opinion toward artificial intelligence and its correlation with personality traits. This research is specifically concerned with the relationship between openness to experience and conscientiousness with respect to trust in robots. Oksanen et al. [15] examined the relationships between personality traits, such as conscientiousness and openness to experience, and trust in robots.

Park and Woo [11] reported that extraversion was associated with negative emotions about AI and low functionality of AI. Agreeableness correlated with both positive and negative emotions toward AI and was linked to higher perceived sociality and functionality of AI. Conscientiousness was associated with fewer negative emotions and higher perceived functionality, but lower perceived sociality. The results indicated that neuroticism was associated with negative emotions about AI and high sociality of AI, whereas openness was positively linked to functionality of AI. A consistent finding was that personal innovativeness in information technology was associated with

positive attitudes towards AI. Kaya et al. [8] additionally emphasised the influence of computer use, knowledge about AI, and AI learning anxiety on shaping attitudes. Schepman and Rodway [7] identified the influence of introversion, conscientiousness, agreeableness, corporate distrust, and general trust on attitudes towards AI. Finally, Sindermann et al. [16] reported that neuroticism was positively associated with fear of AI, whereas age, openness, and agreeableness were linked to acceptance of AI. The collective findings of these studies indicate that a range of personality traits, including extraversion, agreeableness, conscientiousness, neuroticism, and openness, can significantly influence attitudes towards AI.

However, the underlying mechanisms through which personality traits influence attitudes towards AI remain incompletely understood. While prior research has underscored the importance of positive orientation in influencing attitudes towards various domains, its specific mediating role in the relationship between personality traits and attitudes towards AI has yet to be extensively examined.

Positive orientation refers to the fundamental tendency to perceive and pay attention to positive aspects of life, experiences, and the self. Components of positive orientation: self-esteem, optimism and life satisfaction. This dimension underlies positive evaluations and contributes to individuals' well-functioning. A positive orientation is a general tendency to respond positively to life experiences [17].

Research on healthy adult populations has consistently demonstrated that this disposition is systematically related to the Big Five personality traits. In multiple cross-sectional studies, extraversion and emotional stability (low neuroticism) emerge as the strongest positive correlates of positive orientation [18,19]. High extraversion, characterized by sociability, positive affect, and energetic engagement with the environment, supports an optimistic and satisfied view of life. Similarly, emotional stability reduces susceptibility to negative emotions, thereby reinforcing a generally positive outlook. Conscientiousness is also positively associated with positive orientation [19]. Individuals high in conscientiousness tend to set goals, plan effectively, and persevere in the face of challenges—behaviors that contribute to life satisfaction and optimism. Agreeableness correlates with positive orientation through prosocial attitudes, empathy, and cooperative behaviors, which foster supportive social relationships and enhance well-being [19]. The association between openness to experience and positive orientation is more variable, but studies in adult samples have reported positive links [18,19]. This relationship may stem from cognitive flexibility, curiosity, and appreciation for novelty, traits that can help individuals find meaning and satisfaction in diverse life experiences. Overall, findings from healthy adult samples across different cultural contexts indicate that positive orientation is not an isolated construct but is embedded within the broader personality system described by the Big Five. These associations highlight the role of enduring personality traits in shaping a consistently positive perspective on life and oneself [18,19].

A review of the literature on attitudes towards artificial intelligence (AI) reveals a generally positive orientation, with people recognising the potential of AI to solve complex problems and improve society [20]. Nevertheless, this favourable disposition is mitigated by concerns pertaining to privacy, misuse, and security [21]. These concerns are influenced by individual factors such as belief in a dangerous world and technology optimism [21]. The acceptance and fear of AI are key factors in shaping attitudes, with differences observed across industries and organisations [22]. Notwithstanding these reservations, there is a readiness to embrace AI in everyday life, indicating a cautious yet open-minded approach [16].

The study proposed that positive orientation acts as a mediator between personality traits (openness to experience, conscientiousness, extraversion, agreeableness, and neuroticism) and attitudes towards AI. In other words, a positive orientation may explain why some people are more accepting of AI while others remain sceptical. Investigating this mediating effect may provide insights into the underlying psychological mechanisms involved.

## Materials and methods

### Sample

From 15 to 19 December 2023, 1,024 Poles were surveyed, varying in age, gender, education and place of residence and region of Poland. Individuals were interviewed online via the computer-assisted web interview (CAWI) method by the research company Pollster Research Institute (all participants provided written informed consent to take part in the study at the time

of their registration with the research panel), which provided a survey of a representative sample by age, gender, place of residence and education. At the beginning of the analyses, to avoid common method bias affecting the research conclusions, Harman's single-factor test for common methodological bias was used. The results revealed that the variance explained by the first factor was 25.97%, which did not exceed 40%, indicating that there was no serious common method bias. The inter-quartile range of the measured variables was then estimated; on this basis, 102 outlier observations were removed.

The final study sample comprised 922 individuals aged between 18 and 83 years ($M = 48.45$, $SD = 16.82$), with 478 women (51.8%) and 444 men (48.2%). The participants had vocational (26.0%) or high education (28.6%), while only a small proportion had primary education (6.2%). Additionally, 39.2% of the participants had secondary education. A total of 41.6% of the participants came from rural areas, while the others were from towns.

## Measures

The Artificial Intelligence Attitude Scale – 4 items (AIAS-4), developed by Simone Grassini [23] in 2023, is a brief, reliable tool designed to measure general attitudes toward artificial intelligence. It consists of four statements rated on a 1–10 Likert scale, covering beliefs about AI's impact on life, work, future use, and its value for humanity. The overall score is the sum of the four items, with higher scores indicating more positive attitudes. Psychometric analyses show a strong single-factor structure and excellent internal consistency ($α ≈ 0.90$). In the current study, Cronbach's alpha was $α = .93$.

The Ten Item Personality Inventory (TIPI), developed by Samuel D. Gosling, Peter J. Rentfrow, and William B. Swann Jr. [24], is an ultra-brief measure of the Big Five personality dimensions: Extraversion, Agreeableness, Conscientiousness, Neuroticism, and Openness to Experience. Each trait is assessed with two items—one representing the positive pole and one the negative pole—rated on a seven-point Likert scale (1 = strongly disagree; 7 = strongly agree). In this study, the Polish adaptation by Sorokowska et al. [25] was used. The TIPI is designed for contexts where time is limited and a quick personality assessment is needed, such as large-scale surveys or preliminary screenings. Interpretation involves averaging the two items for each trait; higher scores reflect a stronger presence of that trait. Internal consistency is modest due to the brevity of the scale, with Cronbach's α in the Polish version ranging from .45 to .83 across traits.

The Positivity Scale (P Scale), developed by Giovanni Caprara, Maria Alessandra Caprara, and Guido Alessandri, assesses the general tendency to notice and value positive aspects of life, experiences, and oneself. It integrates three psychological components: self-esteem, optimism, and life satisfaction. Respondents rate their agreement with a set of statements on a Likert-type scale, and higher scores indicate a more positive overall life orientation. In this study, the Polish adaptation by Łaguna et al. [26] was used. The scale is applied in personality and well-being research to capture an overarching positive orientation that transcends specific life domains. Internal consistency for the Polish version ranges from Cronbach's $α = 0.77$ to 0.84, with $α = 0.83$ obtained in the present study, indicating high reliability.

## Transparency and openness

The data were accessed on 20 December 2023. The author did not have access to any information that could identify individual participants during or after data collection. All the data have been made publicly available at the Figshare and can be accessed at https://doi.org/10.6084/m9.figshare.29873630. The data were analysed via SPSS version 28 and jamovi 2.5.3. This study's design and analysis were not preregistered.

All procedures performed in studies involving human participants were in accordance with the ethical standards of the institutional research committee and with the 1964 Helsinki Declaration and its later amendments or comparable ethical standards. Informed consent was obtained from all individual adult participants included in the study.

## Results

Pearson's r correlation coefficients were calculated between attitudes towards artificial intelligence and positive orientation, as well as between positive orientation and personality traits (see Table 1).

**Table 1. Correlation between attitudes towards artificial intelligence and positive orientation, personality traits and demographic variables.**

| | M | SD | 2 | 3 | 4 | 5 | 6 | 7 | 8 | 9 | 10 | 11 |
|---|---|---|---|---|---|---|---|---|---|---|---|---|
| 1. Age | 48.45 | 16.82 | .26^ | .28+*** | .15+*** | .00 | .15*** | .19*** | −.15*** | .05 | .13*** | .27*** |
| 2. Gender | | | | .09^ | .02^ | .16^ | .02^ | .01^ | .19^ | .01^ | .08^ | .01^ |
| 3. Place of residence | | | | | −.01+ | −.03+ | .02+ | .07+** | −.05 | .04+ | .03+ | .05+ |
| 4. Education | | | | | | .14+*** | .03+ | .06+* | −.07+** | .10+*** | .04+ | .10+*** |
| 5. Attitude towards artificial intelligence | 22.49 | 8.51 | | | | | .15*** | .06 | −.11*** | .06 | −.03 | .00 |
| 6. Positive orientation | 28.17 | 4.46 | | | | | | .46*** | −.43*** | .07* | .28*** | .33*** |
| 7. Extraversion | 9.58 | 2.64 | | | | | | | −.41*** | .32*** | .36*** | .36*** |
| 8. Neuroticism | 7.62 | 2.73 | | | | | | | | −.02 | −.25*** | −.22*** |
| 9. Openness to experience | 8.59 | 1.83 | | | | | | | | | .13*** | .10*** |
| 10. Agreeableness | 10.61 | 2.12 | | | | | | | | | | .47*** |
| 11. Conscientiousness | 10.84 | 2.41 | | | | | | | | | | |

Annotation. *** $p < .001$; ** $p < .01$; * $p < .05$; ^ - Eta, + - tau-b-Kendalla. Unmarked coefficients represent Pearson's r.

Gender: 0 – female, 1 – male. Place of residence: 1 – village; 2 – town up to 20,000 inhabitants; 3 – town over 20,000 up to 50,000 inhabitants; 4 – town over 50,000 up to 100,000 inhabitants; 5 – city over 100,000 up to 200,000 inhabitants; 6 – city over 200,000 up to 500,000 inhabitants; 7 – city over 500,000 inhabitants. Education: 1 – primary, 2 – vocational, 3 – secondary (high school), 4 – university (tertiary).

The attitude towards artificial intelligence correlated positively with positive orientation ($r = .15$) and negatively with neuroticism ($r = −.11$) – individuals with higher levels of positive orientation and lower levels of neuroticism were more likely to have positive attitudes towards AI. No associations were found between attitudes towards AI and extraversion, openness to experience, agreeableness, or conscientiousness.

A positive orientation has been shown to correlate positively with extraversion ($r = .46$), openness to experience ($r = .07$), agreeableness ($r = .28$), or conscientiousness ($r = .33$) and negatively with neuroticism ($r = −.43$). Individuals with a high positive orientation have been observed to exhibit higher levels of sociability and activity, openness to new experiences, agreeableness and kindness, discipline, and emotional stability.

A bootstrapping mediation analysis was employed to ascertain the role of positive orientation in the relationship between personality traits and attitudes towards artificial intelligence [27,28]. The mediation was calculated via the macro procedure PROCESS for IBM Statistics, version 4.2, and via jamovi 2.5.3. The simple mediation effect (Model 4) was selected, in which the independent (explanatory) variable acting as a predictor (personality traits) is related to the dependent (explained) variable (attitude towards AI) through a third variable acting as a mediator (positive orientation) (Fig 1).

Bootstrapping 5000 with adjusted confidence intervals (95% CI) was employed to estimate the significance of the indirect effects. In accordance with the recommendations, nonstandardised regression coefficients were reported – the values of individual paths [27,28].

Initially, the relationship between extraversion and attitudes towards AI was tested in the context of the mediating role of positive orientation (Table 2).

In the aforementioned model, a substantial indirect effect was obtained (B = .243, SE = .063, 95 CI (0.123, 0.369)), indicating that positive orientation serves as a mediator of the relationship between extraversion and attitudes towards AI. The higher the extraversion is, the greater the positive orientation (a = .78, p < .001). As positive orientation increases, a more favourable attitude toward AI is observed (b = .31; p < .001). The total effect (c = .18) occurs at the level of the statistical trend (p = .09), with an insignificant direct effect (c' = −.06).

In the subsequent model, the relationship between neuroticism and attitudes towards AI was examined within the context of the mediating role of positive orientation (Table 3).

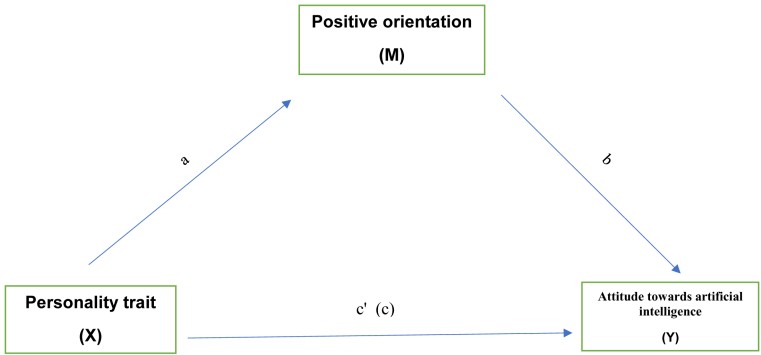

**Fig 1. Model of positive orientation in the relationship between personality traits and attitudes towards artificial intelligence.**

**Table 2. The mediation model of positive orientation in the relationship between extraversion and attitudes towards artificial intelligence.**

| Variables | | Positive orientation | | | | Attitude towards artificial intelligence | | |
|---|---|---|---|---|---|---|---|---|
| | *Path* | *B* | *SE* | *t* | *Path* | *B* | *SE* | *t* |
| Constant | | 20.68 | .49 | 42.21*** | | 14.35 | 1.79 | 8.03*** |
| Extraversion | a | .78 | .05 | 15.88*** | c | .18 | .11 | 1.70 |
| | | | | | c' | −.06 | .12 | −.53 |
| Positive orientation | | | | | b | .31 | .07 | 4.42*** |
| $R^2$ | | 0.22 | | | | .02 | | |
| *F* | | 252.05*** | | | | 11.23*** | | |

The value of the indirect effect was $B = .243$, $SE = .063$, 95% CI (0.123 0.369); ***$p < .001$.

**Table 3. The mediation model of positive orientation in the relationship between neuroticism and attitudes towards artificial intelligence.**

| Variables | | Positive orientation | | | | Attitude towards artificial intelligence | | |
|---|---|---|---|---|---|---|---|---|
| | *Path* | *B* | *SE* | *t* | *Path* | *B* | *SE* | *t* |
| Constant | | 33.54 | .39 | 85.24*** | | 16.84 | 2.45 | 6.87*** |
| Neuroticism | a | −.71 | .05 | −14.51*** | c | −.35 | .10 | −3.40*** |
| | | | | | c' | −.17 | .11 | −1.54 |
| Positive orientation | | | | | b | .25 | .07 | 3.60*** |
| $R^2$ | | 0.19 | | | | .03 | | |
| *F* | | 210.11*** | | | | 12.30*** | | |

The value of the indirect effect was $B = -.174$, $SE = .054$, 95% CI (−0.284, −0.071); ***$p < .001$.

The total mediation effect was obtained, and the total effect was found to be statistically significant (c = −.35; p < .001). However, after the mediator was included, the direct effect became statistically insignificant (c' = −.17). This suggests that a positive orientation completely mitigates the negative relationship between neuroticism and attitudes towards AI. The indirect effect was substantiated through the implementation of the bootstrapping method, where the 95% confidence interval did not include zero (B = −.174, SE = .054, 95% CI (−0.284, −0.071)).

In the subsequent model, the relationships between other personality traits, namely, agreeableness, and attitudes towards AI in the context of the mediating role of positive orientation were examined (see Table 4).

 

**Table 4. The mediation model of positive orientation in the relationship between agreeableness and attitudes towards artificial intelligence.**

| Variables | | Positive orientation | | | | Attitude towards artificial intelligence | | |
| --- | --- | --- | --- | --- | --- | --- | --- | --- |
| | *Path* | *B* | *SE* | *t* | *Path* | *B* | *SE* | *t* |
| Constant | | 22.03 | .72 | 30.50*** | | 16.37 | 2.00 | 8.16*** |
| Agreeableness | a | .58 | .07 | 8.68*** | c | −.12 | .14 | −.87 |
| | | | | | c' | −.31 | .14 | −2.26* |
| Positive orientation | | | | | b | .33 | .06 | 5.16*** |
| $R^2$ | | .08 | | | | .03 | | |
| F | | 75.30*** | | | | 13.71*** | | |

The value of the indirect effect was $B = .193$, $SE = .048$, 95% CI (0.106, 0.290); ***$p < .001$; *$p < .05$

In the aforementioned model, the classic suppression effect was obtained [29], in which the initial relationship between the predictor and the dependent variable is insignificant (total effect) and only after the inclusion of the third variable—the mediator—the direct effect becomes statistically significant (c'=−.31*>c=−.12). The bootstrapping method was employed to ascertain the significance of the suppression effect, and the results indicated a statistically significant relationship (B=.193, SE=.048, 95% CI (0.106, 0.290)).

In the subsequent model, the relationships among another personality trait, conscientiousness, and attitudes toward AI in the context of the mediating role of positive orientation were tested (see Table 5).

The model findings revealed a substantial indirect effect (B=.198, SE=.049, 95% CI (0.106, 0.303)), indicating that positive orientation functions as a mediator in the relationship between conscientiousness and attitudes towards AI. The positive orientation exhibited a strong correlation with conscientiousness (a=.60, p<.001), suggesting that as conscientiousness levels increase, positive orientation also increases. Consequently, as positive orientation increases, so does a more favourable attitude toward AI (b=.33; p<.001).

The significance of the indirect effect in relation to openness to experience and attitudes towards AI in the context of positive orientation was not confirmed (B=.051, SE=.031, 95% CI (−0.005, 0.119)).

## Discussion

The present study examined the mediating role of positive orientation in the relationship between personality traits and attitudes toward artificial intelligence (AI). The findings partially confirmed the proposed model: positive orientation

**Table 5. The mediation model of positive orientation in the relationship between conscientiousness and attitudes towards artificial intelligence.**

| Variables | | Positive orientation | | | | Attitude towards artificial intelligence | | |
| --- | --- | --- | --- | --- | --- | --- | --- | --- |
| | *Path* | *B* | *SE* | *t* | *Path* | *B* | *SE* | *t* |
| Constant | | 21.62 | .64 | 33.85*** | | 15.38 | 1.91 | 8.06*** |
| Conscientiousness | a | .60 | .06 | 10.51*** | c | .00 | .12 | .01 |
| | | | | | c' | −.20 | .12 | −1.62 |
| Positive orientation | | | | | b | .33 | .06 | 4.99*** |
| $R^2$ | | .11 | | | | .03 | | |
| F | | 110.49*** | | | | 12.43*** | | |

The value of the indirect effect was $B = .198$, $SE = .049$, 95% CI (0.106, 0.303); ***$p < .001$.

significantly mediated the effects of extraversion, agreeableness, conscientiousness, and neuroticism on attitudes toward AI, while no mediation was observed for openness to experience. Specifically, higher levels of extraversion and conscientiousness were associated with greater positive orientation, which in turn predicted more favourable attitudes toward AI. Conversely, neuroticism showed a negative indirect effect, indicating that positive orientation buffered the detrimental impact of emotional instability on AI acceptance. Interestingly, the analysis revealed a suppression effect for agreeableness, suggesting a more complex pattern in which positive orientation conceals an underlying negative association between agreeableness and attitudes toward AI.

The findings of this study provide valuable insights into the intricate interplay between personality traits, positive orientation, and attitudes towards artificial intelligence. By examining these relationships, the research highlights the complexities involved in understanding how individual differences shape perceptions and acceptance of advanced technologies. Notably, the identification of positive orientation as a mediating factor emphasises the importance of emotional and cognitive frameworks in determining attitudes towards AI. This suggests that positive orientation, which encompasses traits such as optimism, self-esteem, and overall life satisfaction, plays a critical role in influencing how personality traits affect one's receptiveness to AI. The findings of the study demonstrated a positive correlation between positive orientation and attitudes towards artificial intelligence (AI), whereas neuroticism was negatively correlated. These results are consistent with earlier studies [30]. Individuals with higher levels of positive orientation may demonstrate a greater propensity to adapt to and embrace technological advancements, as optimism and a positive attitude have been shown to mitigate apprehension regarding the unknown [30]. This finding is consistent with the notion that individuals who focus on positive aspects and experiences are more likely to view new technologies such as AI optimistically [25,30]. Their general tendency to look for positive outcomes makes them more receptive to embracing AI and its potential benefits [30]. Conversely, those with a high neuroticism trait, defined by emotional instability and anxiety, have been observed to exhibit scepticism toward innovative technologies, as highlighted in the works of Sindermann et al. [16] and McElroy et al. [31]. This finding aligns with the extant literature suggesting that individuals with elevated levels of neuroticism are more prone to anxiety and exhibit greater uncertainty regarding AI [32].

The absence of substantial correlations between attitudes towards AI and the remaining Big Five personality traits (extraversion, openness to experience, agreeableness, and conscientiousness) indicates that these traits may not exert a considerable influence on attitudes towards AI within this specific context, suggesting that the their influence of these traits may be indirect rather than direct. These results contradict the findings of previous studies [7,11–14]. The observed variations in outcomes may be attributable to discrepancies in research methodologies, variations in participant demographics, or the distinct contextual settings of the studies. The present findings suggest that the influence of personality on attitudes towards AI remains ambiguous and requires further investigation.

The findings of the mediation analyses demonstrate that positive orientation serves as a mediator in the relationship between extraversion and attitudes towards AI. Extraversion, characterised by sociability, assertiveness, and a tendency to seek out stimulation, often predisposes individuals to more favourable views of new experiences, including the adoption of AI technologies. According to the findings of Caprara et al., individuals with higher extraversion, who tend to be more sociable, energetic and optimistic, may benefit from positive orientation as a mechanism that facilitates their acceptance of technology. This finding is consistent with previous research indicating a significant relationship between extraversion and optimism, which promotes adaptation and openness to new experiences [30]. In the case of extraversion, the indirect effect was significant, suggesting that positive orientation fully explains mediates the impact of extraversion on attitudes towards AI.

A similar mediation effect was observed for neuroticism and conscientiousness. In the case of neuroticism, a total mediation effect was obtained, whereby positive orientation completely eliminates the negative impact of neuroticism on attitudes towards AI. This suggests that, in people with higher levels of neuroticism, positive orientation may act as a buffer, increasing their openness to AI. The mediation results confirm that personality traits affect attitudes towards AI mainly through positive orientation rather than directly.

The classical suppression effect of agreeableness is noteworthy, as it suggests that positive orientation plays a crucial mediating role in this relationship. This effect indicates a more complex statistical relationship than simple mediation. In classical mediation, a mediator transmits part or all of the effect of an independent variable on a dependent variable, resulting in a reduction of the direct relationship once the mediator is included. In contrast, suppression occurs when the inclusion of a third variable—here, positive orientation—reveals or strengthens a hidden relationship that was previously obscured. In the present study, the direct relationship between agreeableness and attitudes towards AI was initially nonsignificant but became significant and negative after accounting for positive orientation. This suggests that positive orientation shares variance with agreeableness that conceals their opposing associations with attitudes toward AI. In other words, the presence of a positive orientation highlights the seemingly nonexistent relationship between agreeableness and attitudes towards AI. When positive orientation was taken into account, the direct effect became significant. Notably, the relationship is negative; that is, the higher the agreeableness is, the more negative the attitude towards AI. However, this relationship emerges only in the context of the mediating role of positive orientation. Suppression indicates that a positive orientation 'unmasks' the real impact of agreeableness on attitudes towards AI. This suggests that more compliant individuals may actually have a more negative attitude toward AI, but this relationship is masked by a positive orientation. In the absence of mediation, agreeableness may appear unrelated to attitudes towards AI; however, when positive orientation is considered, a negative relationship becomes evident. Conceptually, this finding implies that although agreeableness is generally linked to prosociality, cooperation, and trust, these characteristics may also predispose individuals to be more cautious or morally concerned about emerging technologies. Positive orientation, encompassing optimism and life satisfaction, may counterbalance this cautious tendency, masking the underlying scepticism associated with higher agreeableness. Once positive orientation is statistically controlled, the more critical or apprehensive aspect of agreeableness becomes visible. This pattern underscores that suppression effects do not simply indicate statistical artefacts but can illuminate opposing psychological mechanisms within personality traits, offering a more nuanced understanding of how prosocial dispositions interact with emotional outlooks in shaping AI acceptance.

Importantly, the indirect effect of openness to experience was not confirmed, suggesting that positive orientation does not play a significant role in the context of attitudes towards AI. This may indicate that openness to experience affects attitudes towards AI in a different way or through different mechanisms. As noted by Stein et al. [9], openness is associated with intellectual curiosity and critical evaluation of new technologies rather than with uniformly positive attitudes. Individuals high in openness may therefore approach AI with greater reflection and ambivalence, balancing interest in innovation with awareness of its potential risks.

For conscientiousness, mediation was also significant, indicating that more conscientious individuals, who are characterised by a greater positive orientation, may be more accepting of AI. Conscientious individuals, known for their organisation, dependability, and disciplined approach, may view AI as a means to increase efficiency and productivity, thereby fostering a positive attitude. Research by Zhang [14] complements this notion, indicating that conscientiousness contributes to greater trust in AI applications in professional settings.

While promoting positive orientation can facilitate openness toward AI, it is equally important to acknowledge the potential risks of overly positive attitudes. Excessive optimism or uncritical acceptance of AI systems may lead individuals to underestimate ethical, privacy, or safety concerns, and to overtrust algorithmic decisions without adequate scrutiny. As noted in prior studies, automation bias and techno-enthusiasm can reduce users' willingness to question AI outputs, thereby increasing the likelihood of errors or unfair outcomes in applied settings [5,6]. Consequently, efforts to enhance AI acceptance should aim to balance optimism with critical awareness, fostering what might be termed 'informed trust'—a stance that combines openness to innovation with careful evaluation of its limitations and societal implications.

The findings indicates that the promotion of a positive orientation could be a strategic approach in efforts to enhance the acceptance of AI technologies across different populations. It is suggested that the fostering of attitudes of a more positive nature more positive emotional tendencies, as exhibited by individuals with higher levels of positive orientation,

could help reduce resistance or apprehension toward AI. Research by Chen & Wen [1] emphasised the potential of education and awareness programs in cultivating positive attitudes toward AI, particularly among demographics that traditionally exhibit scepticism.

Promoting positive orientation offers a practical avenue for enhancing the acceptance of artificial intelligence in society. Drawing on existing psychological literature on positive orientation and well-being [18,33,34], educational initiatives that incorporate elements of positive psychology—such as optimism training, self-efficacy development, and resilience-building—may foster greater openness to technological innovation. Programs aimed at increasing AI literacy could highlight both the potential benefits of AI and the mechanisms that ensure its ethical and safe use, thereby reducing uncertainty and fear. Encouraging individuals to adopt a constructive, future-oriented mindset may strengthen their ability to adapt to technological changes and perceive AI as a source of opportunity rather than a threat.

Furthermore, communication strategies that frame AI as a supportive tool designed to augment rather than replace human capabilities may align with individuals' positive orientation and promote trust. Transparent and balanced messaging—emphasizing both the opportunities and limitations of AI—can reinforce realistic optimism, a concept widely discussed in the literature on adaptive functioning and well-being [33,35]. Drawing on insights from positive psychology and communication research, policymakers, educators, and technology developers can thus design interventions that simultaneously enhance optimism, awareness, and critical reflection—factors essential for fostering balanced and informed acceptance of AI.

The conclusions drawn in the present study are contingent upon the characteristics of the sample and the measurement methods used. By focusing on a singular population from Poland, it may not be possible to recognise the constraints involved in generalising findings to broader, global contexts. Individuals from diverse educational backgrounds may exhibit differing levels of technological apprehension, which could influence their attitudes towards AI [8]. The generalizability of the present findings beyond the Polish context warrants careful consideration. Attitudes toward AI are shaped not only by individual dispositions but also by culturally embedded narratives concerning technology, progress, and human agency. Recent research indicates that cross-cultural variability in AI perceptions is substantial, with trust, moral acceptability, and perceived usefulness differing systematically across societies [36–38]. Cultural frameworks—such as individualism versus collectivism, secularism versus religiosity, or levels of institutional trust—can moderate the relationship between psychological traits and attitudes toward technological innovation [36,39]. In cultures characterised by higher social and institutional trust, positive orientation may act as a catalyst that amplifies openness to AI, whereas in societies marked by ambivalence toward authority or concerns about technological displacement, the same disposition might serve more as an adaptive buffer than a driver of acceptance [37,40]. From this perspective, the Polish results may reflect a sociocultural context of cautious optimism, where technological advancement is recognised as valuable yet approached with ethical reflection and concern for social consequences. Polish surveys indicate that while the public generally supports digitalisation and recognises the benefits of AI, there remains notable scepticism regarding automation and potential threats to employment and privacy [41,42]. Such a climate may have shaped the mediational pattern observed—where positive orientation compensates for underlying uncertainty rather than merely amplifying enthusiasm. Consequently, the current findings may illustrate how culturally specific narratives about AI interact with personality-based dispositions to form nuanced, context-dependent attitudes. Recognising these dynamics opens a path for comparative research that explores whether the observed mediation by positive orientation represents a universal psychological process or a culturally contingent adaptation.

The reliance on self-reported datasets data introduces potential biases, wherein in which participants may alter responses to align with perceived social norms or expectations [13]. This underscores the need for further research employing diverse methods, such as observational or experimental designs, to establish causal relationships with greater robustness. Moreover, the study primarily explored the Big Five personality traits and positive orientation; however, a multitude of other factors contribute to attitudes towards AI, including anxiety about technology and perceptions

of AI's risk versus benefit. Addressing these factors could enhance the proposed frameworks and their predictive power. The use of the Ten Item Personality Inventory (TIPI) entails certain psychometric constraints, particularly with regard to the measurement of openness to experience. Although TIPI offers the advantage of brevity, its reduced number of items may limit the reliability and depth of construct assessment compared to longer, more comprehensive personality inventories. It is important to note that the actual use of artificial intelligence (AI) by the participants was not examined. The study focused on self-reported attitudes and characteristics rather than on direct behavioral measures of AI adoption or interaction. This absence of behavioral data constitutes a clear limitation, as it restricts the ability to draw conclusions about real-world AI use.

Consequently, it is recommended that future research should not only replicate this study with varied samples and settings but also explore how fluctuating socio-technical landscapes interact with these psychological constructs. Such inquiry could spur new hypotheses about the evolving dynamics between personality, psychological orientation, and technological acceptance, enriching the theoretical framework surrounding human-technology interaction.

In conclusion, the present study demonstrates that personality traits, particularly when filtered through the lens of positive orientation, play a meaningful role in shaping attitudes toward artificial intelligence. Positive orientation emerges as a key psychological mechanism that fosters openness to technological change while buffering negative emotional tendencies such as anxiety or distrust. At the same time, the findings highlight the importance of maintaining a balanced perspective—encouraging optimism toward AI while avoiding uncritical acceptance. By integrating individual, emotional factors, this study contributes to a more comprehensive understanding of the psychological foundations of AI acceptance and offers practical insights for education, communication, and policy aimed at promoting responsible and informed engagement with emerging technologies.

## Author contributions

**Conceptualization:** Wiesław Talik.

**Data curation:** Wiesław Talik.

**Formal analysis:** Wiesław Talik.

**Funding acquisition:** Wiesław Talik.

**Investigation:** Wiesław Talik.

**Methodology:** Wiesław Talik.

**Project administration:** Wiesław Talik.

**Resources:** Wiesław Talik.

**Software:** Wiesław Talik.

**Supervision:** Wiesław Talik.

**Validation:** Wiesław Talik.

**Visualization:** Wiesław Talik.

**Writing – original draft:** Wiesław Talik.

**Writing – review & editing:** Wiesław Talik.

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
