## [Decision Letter · Decision Letter 0]

8 Aug 2025

Dear Dr. Talik,

Thank you for submitting your manuscript to PLOS ONE. After careful consideration, we feel that it has merit but does not fully meet PLOS ONE’s publication criteria as it currently stands. Therefore, we invite you to submit a revised version of the manuscript that addresses the points raised during the review process.

We look forward to receiving your revised manuscript.

Kind regards,

Jaroslaw Kozak, Ph.D.

Academic Editor

PLOS ONE

Journal Requirements:

2. Thank you for stating the following financial disclosure: [The work was supported by the Institute of Social Science, The John Paul II Catholic University of Lublin under Grant 1118.].

3. Thank you for uploading your study's underlying data set. Unfortunately, the repository you have noted in your Data Availability statement does not qualify as an acceptable data repository according to PLOS's standards.

Additional Editor Comments:

Dear Author,

Thank you for submitting your manuscript and for your work on this important and timely research topic.

Neither of the reviews received are unequivocally negative; the reviewers appreciate both the importance of the issue raised and the soundness of the statistical analysis. However, their comments are clear: the text in its current form does not yet meet publication standards and requires substantial reworking, both in terms of content and language.

I do not recommend rejecting the manuscript at this stage, but serious substantive and editorial revisions are necessary. Without them, further processing of the text will be very difficult.

Key challenges:

1. Reconstruction and deepening of the discussion

The discussion needs a better structure – key findings (especially mediation hypotheses) should be clearly highlighted at the beginning of this section.

The following should be explained in detail and conceptually:

the lack of mediation for openness – with possible theoretical interpretations,

the suppression effect for conciliatoriness – which is unclear in its current form.

Please also consider the potential risks associated with an overly positive attitude towards AI, as mentioned by both reviewers.

2. Transparency of methods and data

Descriptions of tools must be complete: full name, authors, application, interpretation of results.

Table 1 must include:

mean values and standard deviations,

demographic variables (age, gender, place of residence, education),

full correlation matrix (“all-against-all”).

3. Limitations of the study

The limitations associated with the use of TIPI, especially for openness, should be clearly indicated.

Please add information that the actual use of AI has not been investigated and treat this as a clear limitation.

Please also note the limited possibility of generalizing the results beyond the Polish context.

4. Language and style

The text requires linguistic and editorial revision. There are some complex, unclear sentences and redundancies.

I recommend seeking the assistance of a professional language editor before resubmitting the revised version.

good luck Wiesław! :)

Reviewers' comments:

Reviewer's Responses to Questions

**Comments to the Author**

1. Is the manuscript technically sound, and do the data support the conclusions?

Reviewer #1: Yes

Reviewer #2: Yes

2. Has the statistical analysis been performed appropriately and rigorously?

Reviewer #1: Yes

Reviewer #2: Yes

3. Have the authors made all data underlying the findings in their manuscript fully available?

Reviewer #1: Yes

Reviewer #2: Yes

4. Is the manuscript presented in an intelligible fashion and written in standard English?

Reviewer #1: Yes

Reviewer #2: Yes

Reviewer #1: Please include the full names of the measurement scales in the abstract.

Additionally, please state in the abstract that the hypotheses were only partially confirmed — specifically, the hypothesis regarding openness was not supported.

The sentence "Park and Woo [6] reported that extraversion was associated with negative emotions and low functionality, whereas agreeableness was linked to both positive and negative emotions, as well as high sociality and functionality.” needs to be rephrased, as in its current form it suggests a positive correlation between extraversion and negative emotions. The subsequent sentences should also be clarified to indicate that they refer to attitudes towards AI.

It would be beneficial to include a brief review of studies on the relationship between positive orientation and the Big Five personality traits in the Introduction.

Furthermore, the introduction should begin with several arguments highlighting the potential negative consequences of adverse attitudes towards AI. It would also be valuable to mention examples of negative consequences resulting from overly positive and uncritical attitudes towards AI.

In the Methods section, please start each instrument description with its full name, the authors, and the purpose for which the tool was used in this study. Clearly explain what high scores on each scale indicate.

In Table 1, please add information about the means and standard deviations for each variable.

Also, include sociodemographic variables such as age, gender, place of residence, and education.

Table 1 should be structured to allow for “all-against-all” correlations — that is, all variables should appear both in columns and rows.

Were participants asked whether they use AI, how frequently, and for what purpose? If not, please mention this as a limitation of the current study.

The sentence “In the aforementioned model, a substantial indirect effect was obtained (B = .243, SE = .063, 95CI (0.123, 0.369)), indicating that positive orientation serves as a significant mediator” — please remove the word “significant”, as there is no such thing as a “non-significant mediator.” This correction should be applied throughout the descriptions of analyses.

The Discussion section should begin with the most important results, i.e., those related to the mediation hypothesis.

Please elaborate further on the lack of mediation for openness. Provide additional interpretations — for example, that openness is the personality trait least associated with positive orientation, likely because openness to experience involves openness to both positive and negative experiences. Individuals high in openness may exhibit ambivalent attitudes toward AI — strongly positive in some respects, but also negative, as they may perceive the potential threats associated with AI’s power.

Please also address in the discussion the potential risks of overly positive attitudes toward AI.

Add a section discussing practical implications of the study's findings in psychological practice.

Finally, include a conclusion at the end of the discussion

Reviewer #2: This manuscript reports an investigation that is timely and methodologically rigorous regarding the mediation of positive orientation relative to Big Five personality traits and attitudes toward artificial intelligence (AI). The study is theoretically grounded and statistically strong; it provides significant insights into individual variation in perception of AI, which is increasingly important given the widespread integration of AI into diverse societal sectors.

Use of a large and diverse sample enhances generalizability to the Polish context. Choice of validated instruments including AIAS 4, TIPI PL, and Positivity Scale further corroborates psychometric quality of this research. Statistical approach, especially mediation analysis using bootstrap techniques, is appropriate and executed well and allows meaningful interpretation of indirect effects.

This study reports several significant findings. Positive orientation significantly mediates the relationship among extraversion, conscientiousness and agreeableness via suppression with respect to attitudes toward artificial intelligence (AI). Of particular importance, this mediation attenuates negative impact of neuroticism. Direct effects for openness to experience were also found but are deemed insignificant and cautiously interpreted; undue claims are avoided. Contribution of this paper is conceptual in nature, emphasizing that positive orientation functions as a psychological lens through which attitudes toward emerging technologies are formed.

Some suggestions for the authors of the paper, to improve the quality of the submission.

Clarify the Role of Suppression in Agreeableness:

The finding of a suppression effect involving agreeableness is both statistically and theoretically interesting. However, the explanation of suppression could be expanded with greater conceptual clarity. Currently, the discussion is dense and somewhat circular. A clearer exposition on how suppression differs from full or partial mediation—and its implications for interpreting the negative direct effect—would improve accessibility for a broader readership.

Address TIPI Limitations:

While the use of the Ten Item Personality Inventory (TIPI) is defensible due to space and time constraints in survey research, the limitations of such brief measures (e.g., low internal consistency, particularly for traits like openness) should be acknowledged more explicitly in the limitations section. This is particularly pertinent given the non-significant effects associated with openness to experience.

Broaden Discussion of Cross-Cultural Generalizability:

The discussion section rightly acknowledges the study's national scope (Poland), but the broader implications for cross-cultural variability in AI attitudes could be better elaborated. The sociocultural specificity of AI narratives may influence the mediational pathways observed and should be reflected more critically.

Language and Style Minor Edits:

While the manuscript is generally well-written, a few grammatical and syntactic issues remain. For example, there are occasional redundancies and overly long sentences in the discussion that could be edited for clarity. A professional language edit may enhance readability, particularly for an international audience.

Expand on Practical Implications:

The conclusion alludes to the potential for using positive orientation as a lever for improving AI acceptance. It would be helpful to suggest concrete pathways for doing so, e.g., through education, optimism-based interventions, or communication strategies, drawing on existing psychological literature.

**Do you want your identity to be public for this peer review?** For information about this choice, including consent withdrawal, please see our Privacy Policy

Reviewer #1: No

Reviewer #2: No

---

## [Author Response · Author response to Decision Letter 1]

6 Nov 2025

Dear Editor Jaroslaw Kozak, Ph.D.,

Please find attached my revised manuscript and a separate file with detailed responses to the reviewers’ and editor’s comments.

I appreciate the constructive feedback and hope that the revised version meets the journal’s requirements.

Kind regards,

Wiesław Talik

---

## [Decision Letter · Decision Letter 1]

21 Jan 2026

The mediating role of positive orientation in the relationship between personality traits and attitudes towards artificial intelligence

PONE-D-25-35036R1

Dear Dr. Talik,

We’re pleased to inform you that your manuscript has been judged scientifically suitable for publication and will be formally accepted for publication once it meets all outstanding technical requirements.

Kind regards,

Jaroslaw Kozak, Ph.D.

Academic Editor

PLOS One

Additional Editor Comments (optional):

Reviewers' comments:

Reviewer's Responses to Questions

**Comments to the Author**

Reviewer #2: All comments have been addressed

2. Is the manuscript technically sound, and do the data support the conclusions?

Reviewer #2: Yes

3. Has the statistical analysis been performed appropriately and rigorously?

Reviewer #2: Yes

4. Have the authors made all data underlying the findings in their manuscript fully available?

Reviewer #2: Yes

5. Is the manuscript presented in an intelligible fashion and written in standard English?

Reviewer #2: Yes

Reviewer #2: All the comments were addressed by the authors, and the article is now of a quality standard for acceptance

**Do you want your identity to be public for this peer review?** For information about this choice, including consent withdrawal, please see our Privacy Policy

Reviewer #2: No

---

## [Editor Report · Acceptance letter]

PONE-D-25-35036R1

PLOS One

Dear Dr. Talik,

I'm pleased to inform you that your manuscript has been deemed suitable for publication in PLOS One. Congratulations! Your manuscript is now being handed over to our production team.

Kind regards,

on behalf of

Dr. Frantisek Sudzina

Academic Editor

PLOS One